# Flexible ferroelectric organic crystals

Magdalena Owczarek[1], Karl A. Hujsak[2], Daniel P. Ferris[1], Aleksandrs Prokofjevs[1], Irena Majerz[3], Przemysław Szklarz[4], Huacheng Zhang[1], Amy A. Sarjeant[1], Charlotte L. Stern[1], Ryszard Jakubas[4], Seungbum Hong[5,6], Vinayak P. Dravid[2] & J. Fraser Stoddart[1]

Flexible organic materials possessing useful electrical properties, such as ferroelectricity, are of crucial importance in the engineering of electronic devices. Up until now, however, only ferroelectric polymers have intrinsically met this flexibility requirement, leaving small-molecule organic ferroelectrics with room for improvement. Since both flexibility and ferroelectricity are rare properties on their own, combining them in one crystalline organic material is challenging. Herein, we report that trisubstituted haloimidazoles not only display ferroelectricity and piezoelectricity—the properties that originate from their non-centrosymmetric crystal lattice—but also lend their crystalline mechanical properties to fine-tuning in a controllable manner by disrupting the weak halogen bonds between the molecules. This element of control makes it possible to deliver another unique and highly desirable property, namely crystal flexibility. Moreover, the electrical properties are maintained in the flexible crystals.

[1] Department of Chemistry, Northwestern University, 2145 Sheridan Road, Evanston, Illinois 60208-3113, USA. [2] Department of Materials Science and Engineering, Northwestern University, 2170 Campus Drive, Evanston, Illinois 60208-3113, USA. [3] Department of Analytical Chemistry, Faculty of Pharmacy, Wroclaw Medical University, Borowska 211a, 50-556 Wrocław, Poland. [4] Faculty of Chemistry, University of Wrocław, F. Joliot Curie 14, 50-383 Wrocław, Poland. [5] Materials Science Division, Argonne National Laboratory, Lemont, Illinois 60439, USA. [6] Department of Materials Science and Engineering, KAIST, Daejeon 34141, Korea. Correspondence and requests for materials should be addressed to S.H. (email: hong@anl.gov) or to V.P.D. (email: v-dravid@northwestern.edu) or to J.F.S. (email: stoddart@northwestern.edu).

In 1921, Valasek[1] observed that the spontaneous polarization of Rochelle salt (potassium sodium tartrate tetrahydrate) crystals can be inverted by applying an external electric field. In an attempt to emphasize the conceptual similarity to ferromagnetism, which was already known at the time, the new effect was later named ferroelectricity. In the interim, it has become a workhorse in numerous technologies, finding applications in data and energy storage[2–4]. In crystals, ferro-electricity is observed only in materials adopting polar space groups, and is accompanied by the related phenomena of piezoelectricity and pyroelectricity, further expanding the applications of ferroelectric materials to piezoelectric actuators, sensors, and transducers[5,6]. In contrast, antiferroelectricity can also be observed in the much more common centrosymmetric structures in the form of double hysteresis polarization–electric field profiles consisting of positive and negative field loops when the applied field is sufficiently strong. The sharp change in polarization and volume as a result of applied field or pressure in these systems has the potential to advance technologies in the fields of energy and sensing.

The most widely used inorganic ferroelectric materials such as perovskite barium titanate and lead zirconate titanate[7–9], while showing excellent performance characteristics, are intrinsically limited in their applicability due to the presence of heavy metals. This factor has fuelled the search for more benign substitutes with comparable electric properties. Organic materials have appeared as promising alternatives, and several room-temperature small-molecule organic ferroelectrics and antiferroelectrics have been reported[10–16]. Aside from other desirable properties, organic materials can theoretically also exhibit flexibility, although in practice most organic ferroelectrics and antiferroelectrics are rigid crystalline solids. The most widely applicable organic materials used in flexible devices are ferroelectric polymers[17–19] that, when prepared as thin films, are suitable for both ferroelectric and piezoelectric applications.

While it would be desirable to be able to produce organic ferroelectric/antiferroelectric materials that are both crystalline and flexible, identifying such materials is challenging, since both crystal flexibility and ferroelectricity/antiferroelectricity are rather rare properties in of themselves. Commonly encountered motifs in organic ferroelectrics/antiferroelectrics are chains of hydrogen bonds, where hydrogen bond donors and acceptors switch roles under the influence of an applied electric field. Although the origins of crystal elasticity are poorly understood, the known examples[20–24] suggest that the elastic and plastic properties might correlate with halogen bonding in the crystals. A search of the Cambridge Structural Database[25,26] revealed that some halogenated imidazoles both crystallize in polar space groups and assemble into chains as a result of forming N–H⋯N hydrogen bonds, and, therefore, they appeared to be a suitable starting point for our investigations. Herein we report that suitably designed trisubstituted haloimidazoles display ferroelectricity and piezoelectricity, altogether with the tendency to produce naturally distorted or elastic crystals, with both electrical and mechanical properties arising from distinctly different structural features of these unique molecules.

## Results

**Crystal morphology and structure.** As many as 12 trisubstituted imidazoles (Fig. 1) bearing at least one halogen atom and capable of forming N–H⋯N hydrogen bond chains were synthesized, and their structure-property relationships were explored at both the mechanical and electrical levels. The trisubstituted imidazoles were crystallized at room temperature from $Me_2CO/H_2O$ mixtures by slow evaporation of the more volatile solvent. The resulting crystals were characterized (Supplementary Tables 1 and 2, and Supplementary Figs 1–5) by single-crystal X-ray diffraction, allowing us to identify a group of isostructural compounds 1–8 displaying very different crystal morphologies (Fig. 1). Whereas the symmetrically substituted imidazoles 2, 3 and 5 formed needle-like crystals, the symmetrically chlorinated imidazoles 1 and 4, as well as the unsymmetrically substituted compounds 6, 7, and 8 behaved quite differently. Remarkably, the crystals of these compounds exhibited substantial natural distortions, sometimes appearing as spirals making multiple full turns. Both experimental and theoretical methods were employed in order to understand the observed unusual morphologies.

The crystallographic data reveal that all compounds 1–8 crystallize in the orthorhombic *Ama*2 space group (Laue group *mm*2) with a short *c*-axis of *ca.* 4 Å, a situation which is characteristic of small halogenated aromatic compounds[27,28]. A thorough analysis of their crystal structures reveals highly anisotropic intermolecular interactions which, in previous reports[21,29], have been identified as a distinctive feature of plastic crystals. The crystal packing (Fig. 2) is governed by three types of interactions of different strengths, namely N–H⋯N hydrogen bonds (2.84–2.97 Å) linking imidazole molecules in infinite chains extending in the *a*-axis direction, electrostatic interactions leading to the stacking of the chains one above the other along the *c*-axis to form sheets with the interchain distance between the imidazole rings in the range of 3.79–4.42 Å, and halogen–halogen interactions (Type II, see below) holding these sheets altogether. The anisotropy of the crystal packing forces translates itself to the crystal microstructure. Images obtained by scanning electron microscopy (SEM) reveal (Fig. 3) the layered structures of the crystals of 1 in the dimension perpendicular to the *bc* planes.

**Computational analysis of intermolecular interactions.** Since we suspected that the ability of the trisubstituted haloimidazole crystals to become deformed during growth might arise from

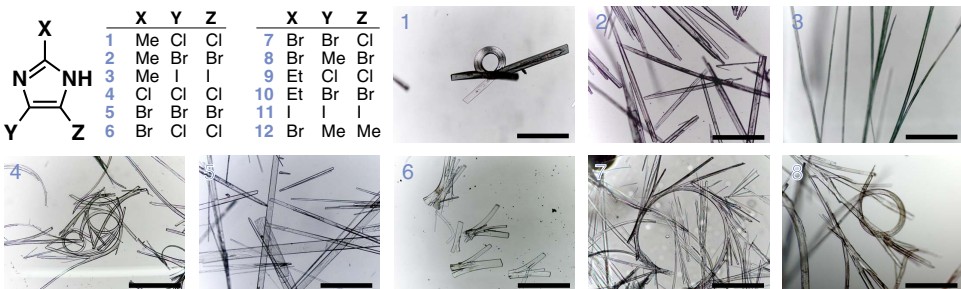

|     | X  | Y  | Z  |     | X  | Y  | Z  |
| --- | -- | -- | -- | --- | -- | -- | -- |
| 1   | Me | Cl | Cl | 7   | Br | Br | Cl |
| 2   | Me | Br | Br | 8   | Br | Me | Br |
| 3   | Me | I  | I  | 9   | Et | Cl | Cl |
| 4   | Cl | Cl | Cl | 10  | Et | Br | Br |
| 5   | Br | Br | Br | 11  | I  | I  | I  |
| 6   | Br | Cl | Cl | 12  | Br | Me | Me |

**Figure 1 | Macroscopic characterization of haloimidazole crystals.** A library of trisubstituted haloimidazoles, **1–12**, from which isostructural compounds **1–8** (optical microscopy images shown) were chosen for investigation of their mechanical and electrical properties. The scale bars are 500 µm.

weak halogen–halogen contacts[21,22,29], these interactions were investigated computationally. Identifying bond critical points in imidazoles **1**, **2** and **4** by Atoms in Molecules (AIM) analysis[30–34] suggests that only halogen–halogen contacts between the

molecules related by a 2-fold screw axis ($m_1$ and $m_4$, and $m_4$ and $m_3$ in Fig. 4a and Supplementary Fig. 6) are essential, and that the strength of the interaction increases with the growth in the atomic radius of the halogen (Supplementary Note 1 and Supplementary Table 3). The geometric parameters of these essential contacts, $\theta_1$ and $\theta_2$ angles (Supplementary Table 4), allow us to classify them as Type II halogen–halogen interactions[35] ($|\theta_1 - \theta_2| \geq 30°$) which, according to the IUPAC definition[36], are true halogen bonds (Supplementary Note 2). To assess the role of these interactions in the structure, the ellipticity ($\varepsilon^{AIM}$) and the dissociation energies ($DE^G$) of the bonds were compared with those in hexachlorobenzene $C_6Cl_6$ (Fig. 4b and Supplementary Table 5), which is known[29] to form plastic crystals that are easily deformable under the influence of external mechanical forces. According to this analysis, the fragment of $C_6Cl_6$ solid-state structure shown in Fig. 4b has only two true halogen bonds, with the rest of halogen–halogen interactions being attributed to close packing. Hence, the weakness of these intermolecular interactions explains (Fig. 4c) the ease with which the stacks of $C_6Cl_6$ molecules can glide past each other to alleviate strain[37], resulting in the plastic deformation of the crystals.

In contrast, the presence of one-dimensional infinite chains composed of imidazole molecules interconnected by hydrogen bonds prevents the gliding between the individual stacks, consistent with the absence of plasticity. In those cases where the halogen bonds are sufficiently weak, as in imidazoles possessing chlorine atoms in positions 4 and 5, the energetic cost of rotation of imidazole rings about the N–H···N bonds

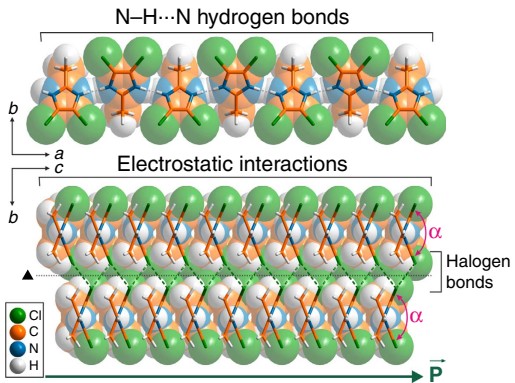

**Figure 2 | Crystal structure of 1–8.** Intermolecular interactions present in the crystal structure of **1** which is used as a representation for all compounds **1–8**. (Top) N–H···N hydrogen bonds linking imidazole molecules in infinite chains extending along the *a*-axis. (Bottom) Crystal packing along the *a*-axis showing both the position of electrostatic interactions as well as halogen bonds. A black dotted line represents the 2-fold screw axis (black triangle). α represents the angle between two consecutive molecules in the N–H···N chain. $\vec{P}$ is a polarization vector.

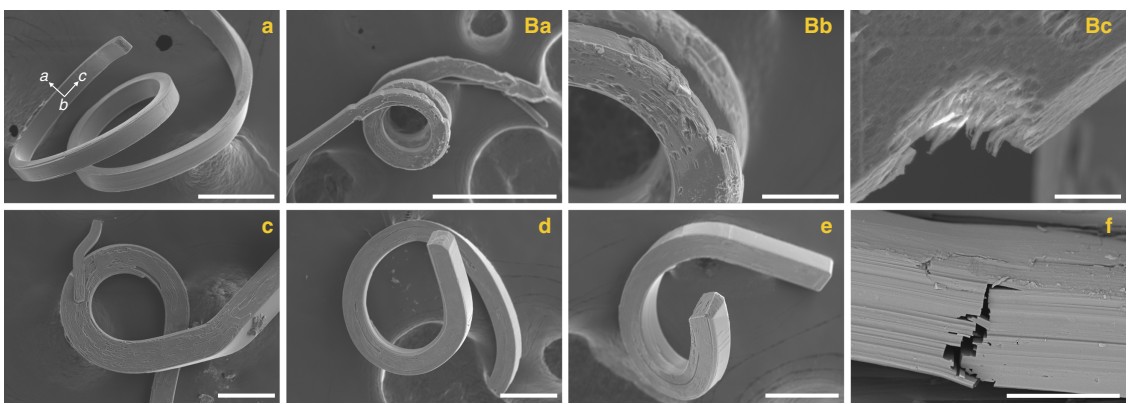

**Figure 3 | Scanning electron microscopy.** SEM images depicting the microstructure of distorted crystals of **1**. The scale bars are 60 (**a**), 300 (**Ba**), 50 (**Bb**), 10 (**Bc**), 100 (**c**), 100 (**d**), 100 (**e**) and 40 μm (**f**).

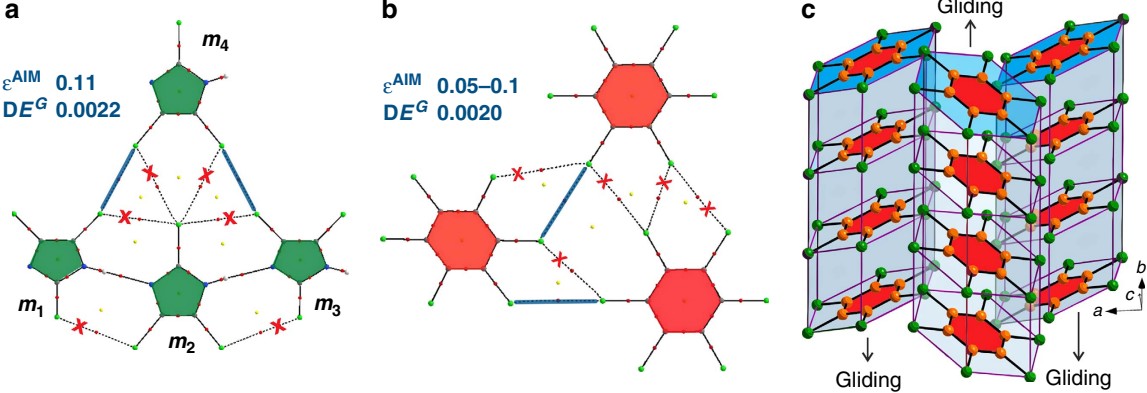

**Figure 4 | Computational analysis of intermolecular interactions.** Representative fragments of the solid-state structures of **4** (**a**) and hexachlorobenzene (**b**) showing bond paths (dashed lines) obtained by AIM analysis. Blue lines denote significant interactions whose existence was confirmed by the AIM method, along with their ellipticity, $\varepsilon^{AIM}$ (a.u.), and dissociation energies, $DE^G$ (kcal mol$^{-1}$), values. Green colour denotes imidazole rings, and red colour denotes benzene rings. (**c**) A schematic representation of the process responsible for the plastic deformation observed for $C_6Cl_6$ crystals.

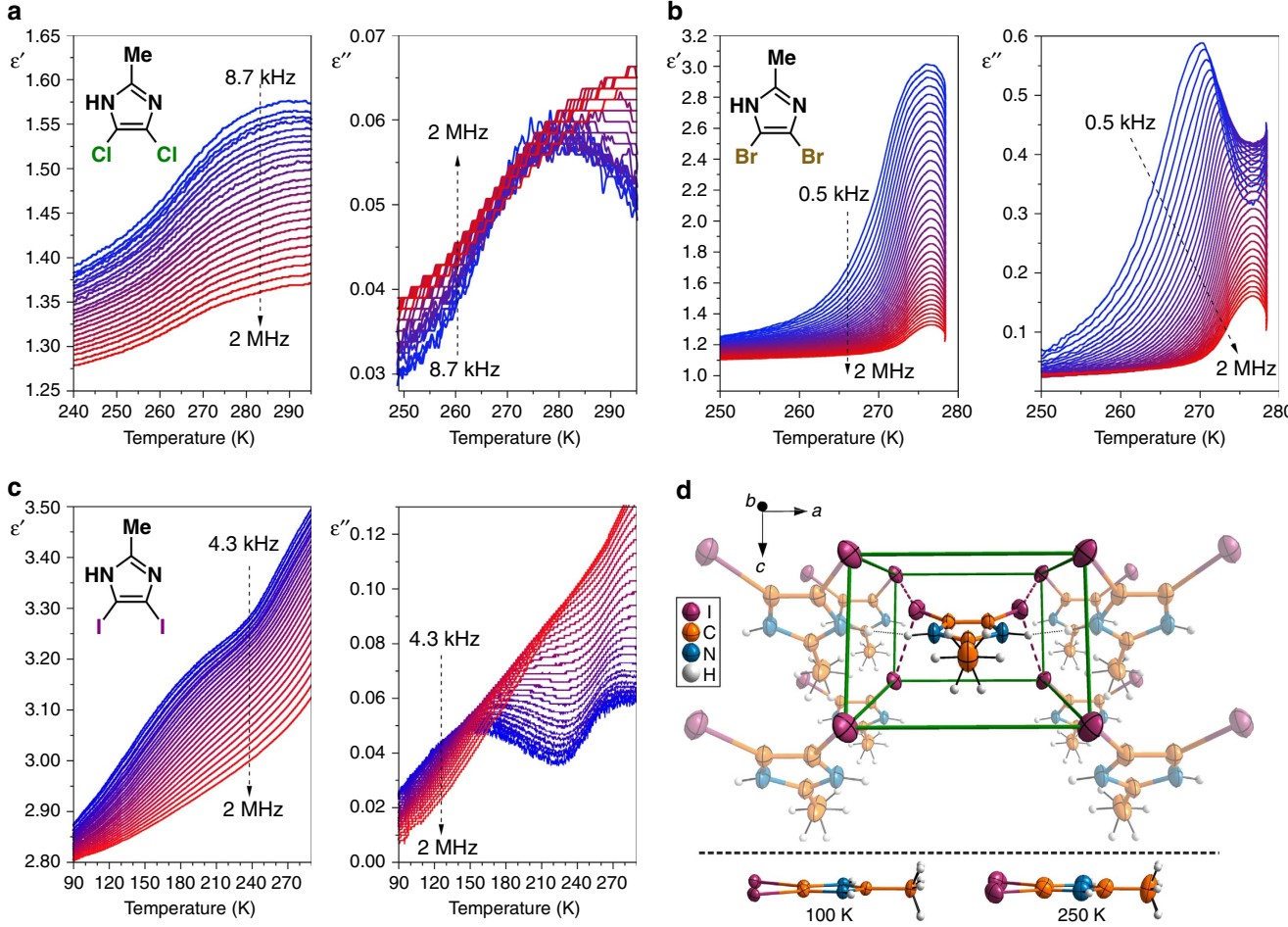

**Figure 5 | Molecular mobility in the solid state.** (**a**–**c**) Temperature dependence of the real, $\varepsilon'(T)$, and imaginary, $\varepsilon''(T)$, parts of the complex electric permittivity measured for **1**, **2** and **3**, respectively. The colour gradient depicts the change in frequency from the lowest (blue) to the highest (red). (**d**) The closest neighbours to a molecule of **3** in the crystal lattice and the comparison of its molecular structure at 100 and 250 K. Intermolecular interactions are marked with dashed lines. Due to the presence of a mirror plane, protons of NH and $CH_3$ groups are disordered between two positions with 0.5 site occupancy factors.

during crystal growth appears to be low, resulting in a change of α dihedral angle (Fig. 2) and a local loss of periodicity. This misalignment, which alternatively can be thought of as the dislocation along the 2-fold screw axis (Fig. 2), induces a slight curvature of the halogen-layered sheets, leading to crystal curving on the macroscopic scale. Similar arguments can be used to rationalize the curved crystal morphology observed in the case of 2,4-dibromo-5-methylimidazole (**8**), where the density of halogen bonds holding the sheets altogether is lower on account of the presence of the 5-Me substituent, even despite the stronger Br···Br bonds. Although other experimental factors, such as temperature and rate of crystallization are known to influence the density of crystalline defects, and are also likely to affect the crystal curvature, they were not investigated.

**Molecular mobility in the solid state**. While the formation of curved crystals appears to indicate the ease with which the haloimidazole molecules can be displaced during the crystal growth, it was also of interest to find out whether the molecular mobility is retained in the solid state. Hence, we performed electric permittivity measurements on a few selected (**1**, **2** and **3**) 4,5-dihalo-2-methylimidazoles. Since the molecules possess permanent dipole moments, the dynamic processes in the crystals

can be expected to reveal themselves under the influence of an alternating electric field, provided sufficient space is available in the crystal lattice. Indeed, relaxation processes were observed (Fig. 5a–c) in the frequency range between 200 Hz and 2 MHz for all three compounds under investigation, as suggested by dispersion and absorption on $\varepsilon'(T)$ and $\varepsilon''(T)$ curves, respectively. The processes are well described by the Cole–Cole or Havriliak–Negami equations[38] (Supplementary Tables 6 and 7), and the parameters derived from these equations allowed us to estimate the activation energies $E_a$, which were found (Supplementary Figs 7–9) to be in the 2.5–53 kcal mol$^{-1}$ range. A plausible rationale for the observed relaxation behaviour in the solid state is that certain molecular motions are not fully restrained by the immediate surroundings (Fig. 5d and Supplementary Note 3) of the haloimidazole molecules. The measurements were performed on pelleted samples containing large numbers of randomly oriented crystals. The calculated activation energies show no simple correlation with the expected strengths of the halogen bonds. Rather, the observed dielectric response should be used as a qualitative proof for the existence of substantial molecular mobility in the crystals. Thus, it appears that, while the curved crystal shape is primarily determined by the strength of the halogen bonds, the dynamic processes in the crystal are independent of their morphology, and were observed in both

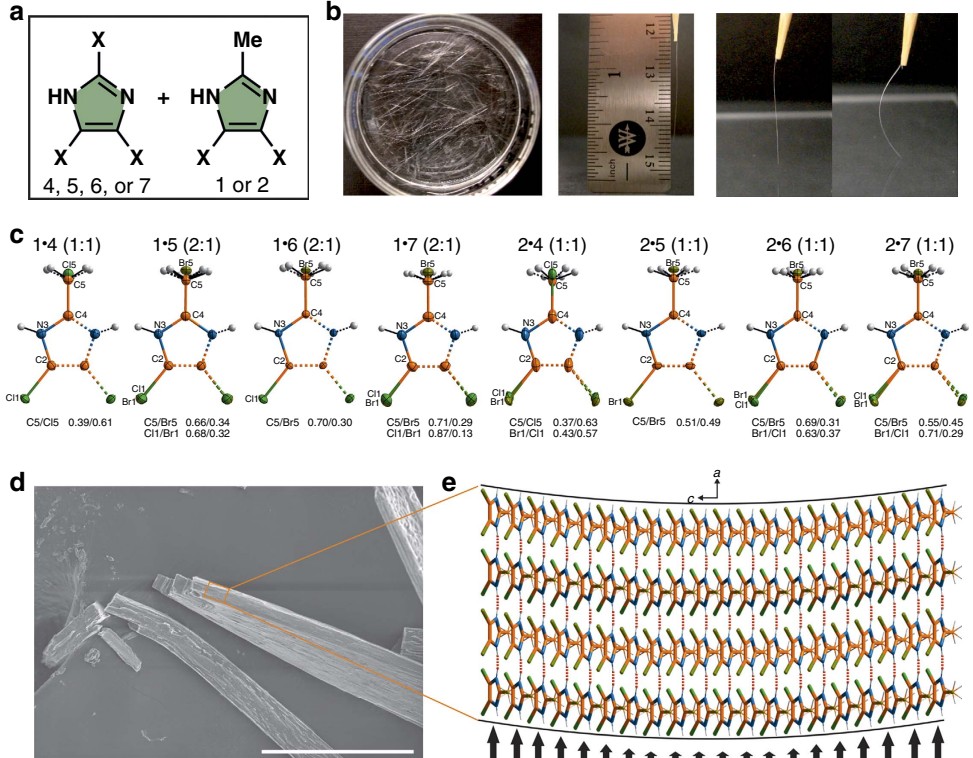

**Figure 6 | Flexibility of haloimidazole mixed crystals.** (**a**) The pattern used for preparing a library of haloimidazole solid solutions. (**b**) Crystalline needles of a solid solution grown from a $Me_2CO/H_2O$ mixture and their elastic properties. (**c**) Asymmetric parts of the mixed crystals unit cell marked with solid lines and symmetrical equivalents marked with dotted lines. Stoichiometric ratios in which components were mixed are presented in parentheses. Occupancy factors of disordered atoms are also given. Thermal ellipsoids are shown at 50% probability. (**d**) SEM images of flexible solid solution crystals and (**e**) the mechanism of a crystal deformation. X···X contacts are not shown for the sake of clarity. The scale bar in **d** is 100 μm.

curved (**1**) and regular (**2** and **3**) crystals. This realization leads us to the conclusion that the possible electrical properties (see below) arising from the non-centrosymmetric crystal structure of these haloimidazole systems should persist regardless of the morphological effects that dictate the crystal shape.

**Flexible mixed crystals.** The pursuit of more dynamic morphological properties led us to explore mixed crystal systems. The halogen bonding influence on crystal shape, as well as the molecular mobility of haloimidazoles in the solid state, appeared to us to be convenient handles for tuning the morphology and mechanical properties of the crystals. We proposed that the halogen bonded network of trihaloimidazoles **4–7** could be disturbed in a controlled manner with the Me groups of 4,5-dihalo-2-methylimidazoles **1** and **2** through co-crystallization (Fig. 6a and Supplementary Table 8), preventing crystal distortion during growth. Indeed, with some tuning of the component ratio, co-crystallization resulted in the formation of 1–4 cm long straight needles (Fig. 6b), even in those cases where individual compounds (**1**, **4**, and **7**) gave curved crystals. According to single-crystal X-ray diffraction (Fig. 6c, Supplementary Figs 10 and 11, and Supplementary Table 9), the samples prepared by co-crystallization are not true co-crystals, but rather are mixed crystals (solid solutions) maintaining the ratio of components used for crystallization. As an additional proof of the structural composition of the mixed crystals (**1•5**, **1•6**, **1•7**, **2•5**, **2•6**, **2•7**), and the consistency of the imidazole ratio throughout the crystallization batch, we performed a series of HPLC experiments (Supplementary Fig. 12, Supplementary Tables 10–15) with five

randomly selected crystals from each batch. Perhaps the most significant attribute of these mixed crystals is the fact that aside from their different morphologies, they were also found to be substantially more flexible than the crystals of pure imidazoles (Fig. 6b, Supplementary Videos S1–S3, and Supplementary Fig. 13). Thus, when taken out of the crystallization dish using a toothpick, and pressed against a glass slide, the crystals can be bent many times without breaking, provided that the applied force does not exceed a certain limit. The elastic properties are most visible to the naked eye if the crystal thickness does not exceed 50 μm, and under optimum circumstances the opposite ends of the elastic needle can be forced to touch without breaking the crystals. The solid solutions were analysed with a nanoindentation technique, and determined to be gel-like from their measured hardness, supporting the flexibility observations (Supplementary Table 16). To assess the extent of the structural changes occurring in elastic crystals, we performed an X-ray diffraction study of a crystal that had been exposed to multiple bending/relaxation cycles. Since no elongation of X-ray diffraction peaks was observed, and the peaks could be indexed to the unit cell identical to that of the crystal before bending, it can be assumed that the molecules regain their original positions in the crystal lattice upon withdrawal of the applied bending force.

On the basis of crystal indexing data and SEM imaging (Fig. 6d), we were able to propose a mechanism for the observed crystal flexibility. Attaching bent elastic crystals to carbon tape used as the SEM substrate, followed by fracturing the mechanically strained crystals, allowed us to acquire SEM images clearly demonstrating the exposed layers in the crystal (Supplementary Fig. 14) corresponding to the N–H···N chains expanding in the

*a*-axis direction. Thus, it seems likely that it is the facile slippage of the imidazole chains past each other in the *ab* plane—distorting electrostatic interactions between imidazole rings in the *c*-axis direction—that confers flexibility upon the mixed crystals (Fig. 6e). The ability of the crystals to deform elastically in response to the bending force appears to be a function of the X···X halogen bonds wherein the potential wells seems to be wide enough to allow the bonds become strained/relaxed without being broken. When the external force is removed, the bonds regain their original position, causing the crystal to bounce back and adopt its original shape.

**Piezoelectric and ferroelectric properties**. Initial electromechanical experiments (Fig. 7) for determining the piezoelectric nature of the materials were carried out using Piezoresponse Force Microscopy (PFM) as the method of choice to characterize the electromechanical behaviour of these systems while simultaneously defining the topographical orientation of the substrate. The oscillation of the conductive cantilever in contact with a piezoelectric system is captured by both the amplitude and the phase of the collective material and the cantilever system. While the amplitude allows us to examine the magnitude of the piezoelectric response, the phase gives us information on the sequence

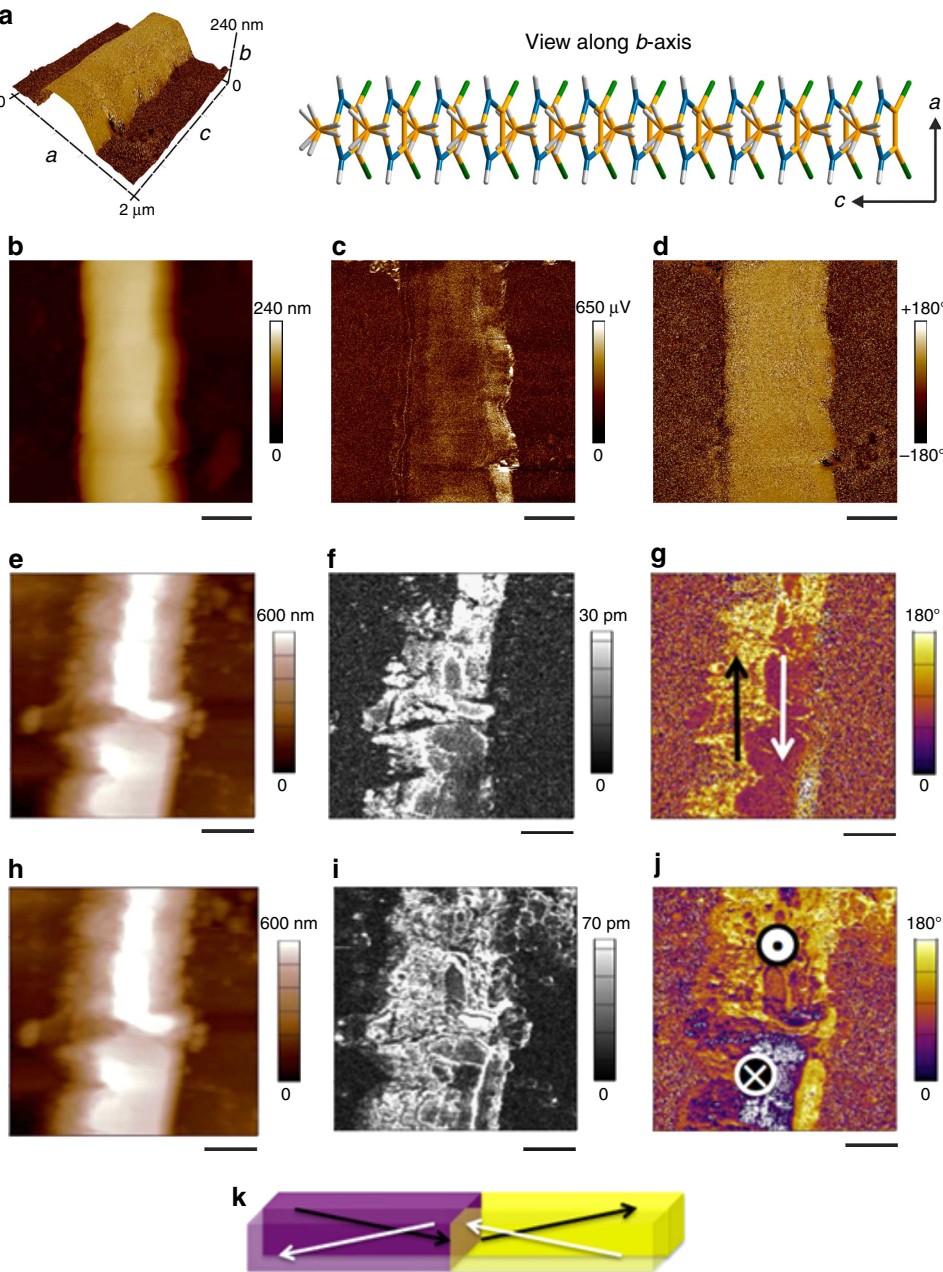

**Figure 7 | Piezoelectric properties and polarization domain structure of 2.** Piezoresponse Force Microscopy (PFM) images (**a–j**) and schematic representation of the domain structure (**k**): (**a**) 3D image of the topography in **b** with the colourmap of the long axis lateral phase (**d**) demonstrating the one-to-one correspondence, and the molecular arrangement in the crystallographic *ac* plane. (**c**) Lateral amplitude of the long axis of the crystal. (**d**) Lateral phase of the long axis response. (**e**) Topography image of the large crystal of **2**. (**f**) Lateral amplitude of the long axis of the crystal. (**g**) Lateral phase of the long axis response. (**h**) Topography image of the same crystalline needle. Vertical amplitude (**i**) and vertical phase (**j**) response of the crystal. (**k**) 3D schematic representation of the domain structure in the crystal imaged by PFM. The scale bars in **b–d** are 470 nm and 1 μm in **e–j**.

of the mechanical oscillation that is related to the orientation of the polarization, allowing us to identify domain structures. More specifically, the piezoelectric coefficient, $d$, is related linearly[39] to the spontaneous polarization, $P_s$, permittivity, $\varepsilon$, electrostrictive coefficient, $Q$, by equation (1)

$$d = 2\varepsilon Q P_s \qquad (1)$$

As ferroelectricity is defined by the switchable polarization under the influence of an external electric field, PFM hysteresis loop measurement is a tool that could probe the ferroelectricity if all the potential artefacts are taken care of. Since the cantilever has only two degrees of freedom—vertical bending and lateral torsion—with which to probe the piezoelectric motion of the sample, physical alignment of either the long or short in-plane axes of the crystalline needle with the torsion of the cantilever, was performed by rotating the sample through 90 degrees in order to obtain the alternate axis.

Since the crystal face indexing of **1–8** and solid solutions of these molecules differs in regards to the position of the crystallographic $a$ and $b$ axes (Supplementary Figs 5 and 11), we investigated the electromechanical properties of two types of crystals: one with the widest (010) face (crystals of **2**) and the other with the widest (100) face (mixed crystals **2•7** and crystals of **8**). Figure 7a shows the three-dimensional (3D) topography image of the analysed crystal of **2** (Fig. 7b) along with crystallographic directions obtained from the X-ray crystal face indexing. As can be observed, there is a strong amplitude response along the long axis of the crystal (Fig. 7c) and a uniform lateral phase (Fig. 7d). The piezoelectric response along the short axis (Supplementary Fig. 15) is effectively zero, in contrast to the in-plane long axis which is aligned with the primary crystallographic polar axis ($c$-axis in $Ama2$ space group). By ramping the applied drive amplitude and monitoring the response of the material, the piezoelectric properties were probed (Supplementary Fig. 16) quantitatively. A linear piezoelectric response is obtained predominantly along the long axis, with a slight piezoelectric response along the vertical dimension as well. Accounting for the background signal, calibrating the vertical deflection signal, and estimating the lateral deflection sensitivity (as described in

the Supplementary Note 4, Supplementary Figs 17–20 and Supplementary Table 17), the vertical piezoelectric and long axis lateral effective piezoelectric coefficients for **2** were roughly estimated to be 0.12 and 2.6 pm V$^{-1}$, respectively; however, these values may be underestimated as a result of unavoidable friction and slip between the cantilever and the sample[40,41]. When larger crystals of **2** were analysed, we started to see polarization domain structures with opposite directions as shown in Fig. 7e–k. Both out-of-plane and in-plane piezoresponse, however, did not show any polarization switching characteristics, indicative of piezoelectricity without ferroelectricity or ferroelectric polarization pinned by charged defects.

Interestingly, this situation changed when the crystals with the crystallographic $a$-axis perpendicular to the widest face were probed. The molecular arrangement along the $a$ direction is presented in Fig. 2 (bottom). For mixed crystals of **2•7**, we were able to switch the lateral piezoresponse by applying a vertical field, which implies polarization switching by external electric field (Fig. 8a–g). This strongly suggests that sample **2•7** possesses ferroelectric properties along the polar axis ($c$-axis) of the crystal. Figure 8f,g and Supplementary Fig. 21 show the vertical and lateral piezoresponse hysteresis loops measured using both pulse and continuous modes (or 'on' and 'off' modes), respectively[42]. The cantilever–sample interaction is strong (Supplementary Fig. 21), even though we used stiff cantilevers with high aspect ratio tip, probably due to the large voltage that was applied in an attempt to switch the polarization. In such cases, hysteresis loops obtained by pulse dc mode provide a clearer picture for identifying switchable polarization. Figure 8f,g presents such loops and shows that only the lateral piezoresponse can be switched from one direction to the other upon application of the external electric field, whereas the vertical piezoresponse cannot be switched within the voltage range of 200 V.

Both vertical and lateral PFM were used to confirm the presence of the ferroelectric and piezoelectric properties in single-component crystals of **8**, which exhibit the same axes orientation as **2•7**. The crystals of **8** exhibited an even weaker vertical piezoresponse than sample **2•7**, indicating greater alignment of the polarization along the long axis of the crystal. As shown in

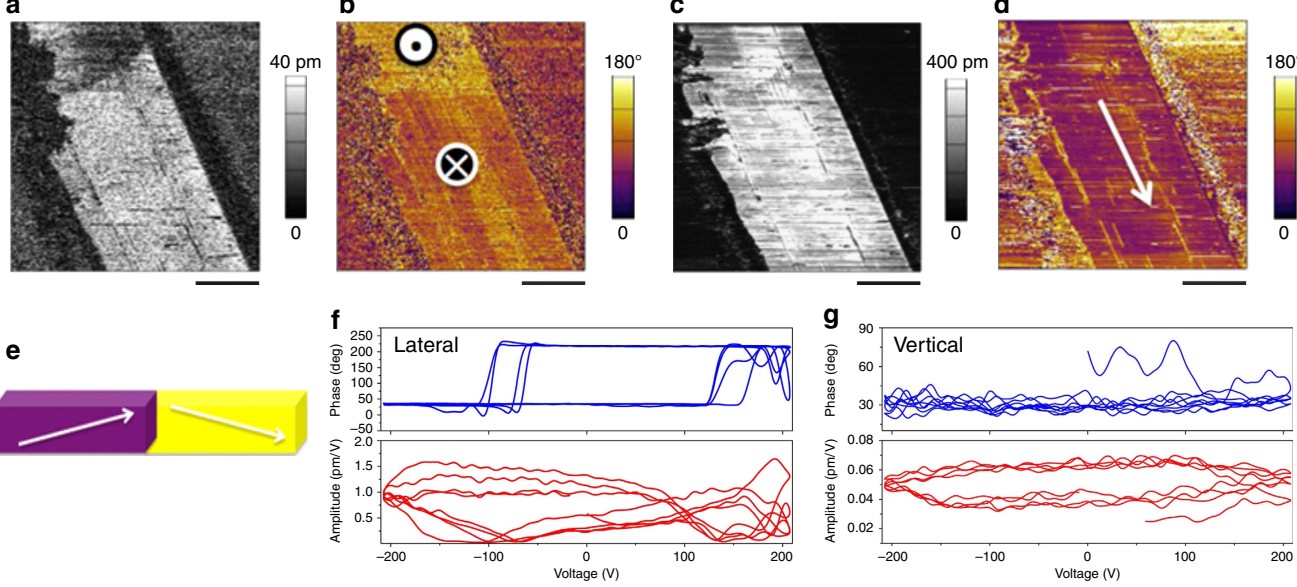

**Figure 8 | Polarization domain and piezoresponse hysteresis loops of 2•7.** Piezoresponse Force Microscopy (PFM) images (**a**–**d**) and schematic representation of the domain structure (**e**): (**a**) Vertical amplitude and (**b**) phase of the crystal. (**c**) Lateral amplitude and (**d**) phase of the long axis response of the crystal. (**e**) 3D schematic representation of the domain structure in the crystal imaged by PFM. (**f**) Lateral amplitude and phase and (**g**) vertical amplitude and phase hysteresis loops obtained by a pulse dc mode. The scale bars in **a**–**d** are 10 μm.

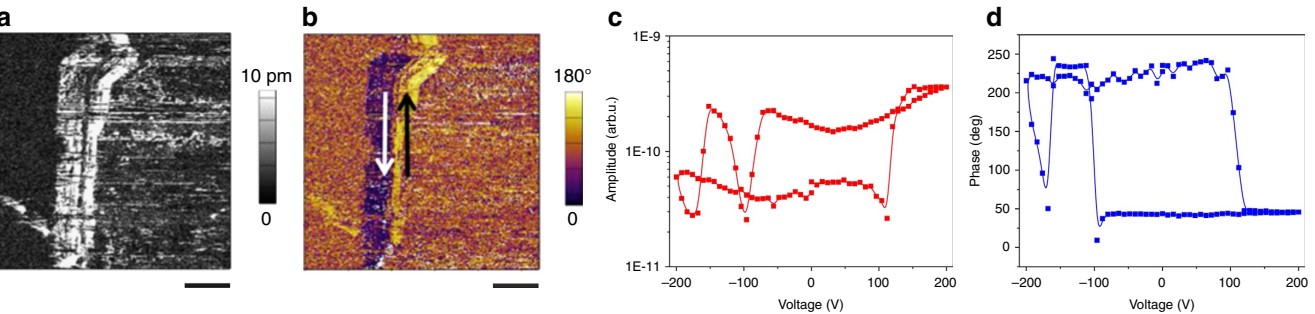

**Figure 9 | PFM images and piezoresponse hysteresis loops of 8.** (**a**) Lateral amplitude and (**b**) phase images of the crystal. (**c**) Lateral piezoresponse amplitude and (**d**) phase hysteresis loops obtained by a pulse dc mode. The scale bars in **a** and **b** are 5 µm.

Fig. 9a,b, the lateral PFM images showed a similar in-plane polarization domain structure to sample **2**. In addition, the piezoresponse hysteresis loop measurement indicated switchable in-plane polarization (Fig. 9c,d) by external electric field in **8**, similar to sample **2•7**, as well as prototypical polymer ferro-electrics[19]. Although the mobility of molecules about the N–H⋯N hydrogen bonds—that is postulated in this work as an explanation for both the natural distortion of crystals and the observed relaxation processes—might also explain the ease of the lateral piezoresponse switching when an external field is applied along N–H⋯N chains, the precise molecular mechanism of the switching still needs to be evaluated in greater detail.

## Discussion

Haloimidazoles constitute a rare case of room-temperature ferroelectric materials on account of their non-centrosymmetric crystal lattice with the polar direction orthogonal to the direction of the N–H⋯N chains. The tendency of haloimidazoles to produce naturally distorted crystals has its roots in the imbalance of crystal packing forces, with halogen bonding playing a substantial role. While distorted crystals have surely been observed previously, they may have been viewed as inferior for crystallographic purposes, and, therefore, were not studied systematically. The halogen bond networks in haloimidazoles can be disrupted in a controllable way by the addition of halogenated 2-methylimidazoles, producing flexible crystalline organic ferroelectrics.

Related benzimidazole systems have previously been shown to possess either ferroelectric or antiferroelectric properties due to mobility of hydrogen atoms in N–H⋯N bridges under the influence of an applied electric field. Therefore, it is conceivable that the haloimidazoles presented in this study may also exhibit antiferroelectric properties in the direction perpendicular to the polar axis. Although several attempts were made to explore this intriguing possibility, no conclusive evidence has been obtained in this regard, which warrants further investigation.

The unique combination of electrical and mechanical proper-ties, enhanced by the structural simplicity of haloimidazole molecules, clearly demonstrates that organic materials may provide alternative sources of piezoelectrics and ferroelectrics. The attractiveness of the haloimidazoles includes their ease of synthesis, coupled with the ability to make simple structural modifications so as to perturb the hydrogen bond chains and weak halogen networks governing the physical properties of these crystalline systems.

## Methods

**Synthesis and crystallization.** Compounds **1** and **2** were purchased from com-mercial suppliers, and were used as received. Syntheses of compounds **3–12** were performed according to previously reported procedures, or their adaptations

(see Supplementary Methods for more details). Crystals and mixed crystals of haloimidazoles were grown from $Me_2CO/H_2O$ solutions by slow evaporation of the more volatile solvent ($Me_2CO$).

**Microscopy.** The optical images were acquired using an Olympus BX53 upright microscope. Scanning Electron Microscopy (SEM) images were collected on a Hitachi S-3400N-II SEM microscope at Northwestern University's EPIC/NUANCE facility and on a Hitachi S-3400N SEM at the Laboratory of Electron Microscopy at University of Wroclaw (Poland). Accelerating voltage of 4–8 kV, and a secondary or backscattered electrons detector (SE or BSE) were used. Samples were coated with Au/Pd to ∼5 nm thickness using a Denton Desk IV Sputter Coater or a Cressington 108A Gold Evaporator before imaging.

**Single-crystal X-ray diffraction measurements.** X-Ray quality crystals of haloimidazoles **1–10** were grown from $H_2O/Me_2CO$ solutions by slow evaporation. Single crystals of **11** and **11 • MeOH** were obtained by slow evaporation of MeCN and MeOH solutions, respectively. Solid solutions were prepared by dissolving the two components in a mixture of $H_2O$ and $Me_2CO$ in a 5 cm Petri dish and evaporating the solvents at room temperature. The amount of materials used for each solid solution is listed in Supplementary Table 8. The ratio and the rate of crystallization were optimized so that elastic needle-shape crystals (∼50 µm in width) were obtained. Diffraction measurements were performed on Bruker Kappa APEX 2 diffractometers (ω and φ scans), equipped with MoKα sealed tube with Triumph monochromator or CuKα microsource and CCD area detector, or Xcalibur diffractometer with graphite monochromated MoKα radiation and CCD area detector. Using Olex2 (Supplementary reference 17), the structure was solved with the ShelXS (Supplementary reference 18) structure solution program using direct methods and refined with ShelXL (Supplementary reference 18) with ani-sotropic thermal parameters for non-H atoms. All H atoms were treated as riding and placed in geometrically optimized positions. H atoms of NH groups in **11** were refined with 0.5 site occupancy factors as they could not be located from the molecular geometry and in the difference Fourier map. In the case of mixed crystals, the initial refinements included C4–C5 distance restrain (1.5 Å), but it was removed during the final stages of the refinement. The crystal data, altogether with experimental and refinement details, are given in Supplementary Tables 1, 2, and 9. Crystallographic data for the structures have been deposited with the Cambridge Crystallographic Data Centre. These data can be obtained free of charge via via www.ccdc.cam.ac.uk/getstructures website. Although the crystal structures of **2** and **12** have been already published (Supplementary reference 1), the previously reported crystal data of both compounds are shown in Supplementary Tables 1 and 2 for the sake of completeness. The asymmetric parts of crystals **1–12** and the mixed crystals are presented in Supplementary Figs 4 and 10, respectively.

**Theoretical calculations.** Wave function files (wfn) of crystal structures were generated at the B3LYP/6-311++G(d,p) level of theory using the Gaussian 09 program, and AIM analyses were performed with the AIMAll program. In comparison with the actual crystal structure of haloimidazoles where the hydrogen atom is disordered between two sites in the N–H⋯N bridge, the position of the NH hydrogen atom was fixed so that it belongs to only one nitrogen atom of the imidazole molecule. This procedure did not influence significantly the parameters of the bond critical point (BCP) of intermolecular X⋯X interactions. See Supplementary Methods for more details.

**Electric permittivity measurements.** The frequency dependence of the complex electric permittivity, $\varepsilon^* = \varepsilon' - i\varepsilon''$, was measured between 90 and 300 K with an Agilent E4980A Precision LCR Metre in the frequency range 200 Hz to 2 MHz. The samples were ground using a mortar and pestle and the powdered materials were pelleted using 150–180 bar pressure. Copper conductive tape was applied onto opposite faces of each pelleted sample to run permittivity experiments. The overall

error in electric permittivity measurements was less than 5%. In the case of **1** and **3**, the response can be described well by the Cole–Cole relation (equation 2) where $\varepsilon_0$ and $\varepsilon_\infty$ are low and high frequency limits of the electric permittivity, respectively, $\omega$ is an angular frequency, $\tau$ is a macroscopic relaxation time, and $\alpha$ is an exponent describing the broadness of the relaxation time distribution. The Cole–Cole plots at several temperatures were fitted to the experimental values which allowed us to determine the parameters ($\varepsilon_0, \varepsilon_\infty, \alpha$ and $\tau$) of equation 2 (Supplementary Table 6). In the case of **2**, the Havriliak–Negami equation (equation 3), with an additional $\beta$ parameter introducing asymmetry to the $\varepsilon''(\varepsilon')$ plot, was used to fit the data (Supplementary Table 7). Activation energy values, $E_a$, were estimated from the Arrhenius relation for the macroscopic relaxation time, equation 4, where $R$ is gas constant, $T$ is temperature, and $C$ is a pre-exponential factor.

$$\varepsilon^*(\omega) = \varepsilon_\infty + \frac{\varepsilon_0 - \varepsilon_\infty}{1 + (i\omega\tau)^{1-\alpha}} \qquad (2)$$

$$\varepsilon^*(\omega) = \varepsilon_\infty + \frac{\varepsilon_0 - \varepsilon_\infty}{[1 + (i\omega\tau)^{1-\alpha}]^\beta} \qquad (3)$$

$$\tau = C \exp\left(\frac{E_a}{RT}\right) \qquad (4)$$

**Piezoresponse force microscopy.** A Bruker Dimension ICON Atomic Force Microscope (AFM), equipped with a Nanoscope V controller in PFM Vertical/Lateral Low Frequency Mode with SCM-PIT cantilevers of nominal spring constant $2.8\,N\,m^{-1}$, was used for piezoelectric measurements. Experiments were performed in the SPID Center of the Northwestern University Atomic and Nanoscale Characterization Center (NUANCE). All images were taken during one imaging session with the same cantilever and laser position in order to allow for quantitative comparison across the samples investigated, without the need to adjust deflection sensitivity or calibrate the phase response of the PFM signal. Imaging was performed in contact mode at a vertical deflection setpoint of 10 nm, a scan rate of 0.1 Hz, and captured at $512 \times 512$ lines. Voltage ramps were performed at a rate of 0.01 Hz and sampled at 1,024 points. An AC voltage of 10 Vpp (peak-to-peak) was applied in both calibration and imaging steps in order to ensure beyond any doubt reliable comparison between samples. For the piezoresponse force microscopy imaging and hysteresis loop measurement, presented in Fig. 7e–j, Fig. 8a–g, and Fig. 9a–d, MFP-3D (Asylum Research, Oxford Instrument) was used with PPP-EFM (Nanosensors, $1.7$–$2.5\,N\,m^{-1}$) at Argonne National Laboratory. We used dual ac resonance tracking (DART) mode for both vertical and lateral PFM imaging as well as piezoresponse hysteresis loop measurements. The drive voltage was 5 V and drive frequencies were 350 and 655 kHz for vertical and lateral PFM experiments, respectively. Crystals for PFM measurements were grown from $MeOH/H_2O$ or $Me_2CO/H_2O$ solutions on a glass slide coated with an Au/Pd layer $\sim 15$ nm thick.

**Nanoindentation tests.** Mechanical properties of the elastic solid solution crystals were measured at ambient temperature using a Hysitron Triboindenter TI 950 system at Northwestern University's EPIC/NUANCE facility. A Berkovitch diamond indenter tip was used to assess the hardness and elastic moduli of the crystals. A trapezoid load-unload function was used for each indent spot: 10 s linear loading and 10 s unloading segments with a dwelling of 6 s at the peak load. The maximum load was applied to all samples at a depth of 250 nm. The final values of hardness and elastic modulus (Supplementary Table 16) represent an average of five indentations performed in different spots on the same material.

**Data availability.** X-Ray crystallographic data have been deposited with the Cambridge Crystallographic Data Centre, CCDC nos 1425308–1425327, and can be obtained free of charge from the Centre via its website (www.ccdc.cam.ac.uk/getstructures). All other data supporting the findings of this study are available within the article and its Supplementary Information.

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

## Acknowledgements

This research (project no. 3) was conducted as part of the Joint Center of Excellence in Integrated Nanosystems (JCIN) at King Abdullaziz City for Science and Technology (KACST) and Northwestern University. We would like to thank both KACST and Northwestern University (NU) for their continued support of this research. The Integrated Molecular Structure Education and Research Center (IMSERC) at NU is acknowledged for the use of their facilities. This work made use of the EPIC facility of the NU*ANCE* Center at Northwestern University, which has received support from the Soft and Hybrid Nanotechnology Experimental (SHyNE) Resource (NSF NNCI-1542205); the MRSEC program (NSF DMR-1121262) at the Materials Research Center; the International Institute for Nanotechnology (IIN); the Keck Foundation; and the State of Illinois, through the IIN. Work at Argonne (S.H., PFM imaging/hysteresis loop measurement and analysis) was supported by the U.S. Department of Energy (DOE), Office of Science, Office of Basic Energy Sciences (BES), Division of Materials Sciences and Engineering. We thank the Wroclaw Center for Networking and Supercomputing for the computer time.

## Author contributions

M.O. conceived the idea and developed the project. M.O., D.P.F. and H.Z. synthesized the compounds studied in this work; M.O. grew crystals and took optical microscopy images; R.J. carried out DSC measurements; S.H. and K.A.H. performed piezoelectric and ferroelectric characterization of the materials; M.O. and A.P. carried out HPLC experiments; M.O. and D.P.F. performed nanoindentation experiments. SEM images were taken by M.O., D.P.F and P.S. Crystal diffraction data were collected by A.A.S., C.L.S. and P.S.; structure determination and refinement were performed by M.O. and P.S. with A.A.S. and C.L.S. assistance. I.M. performed AIM calculations and helped with the AIM data analysis. R.J. and V.P.D. offered intellectual input. M.O., A.P., K.A.H., D.P.F., S.H. and J.F.S. wrote the manuscript with input from all co-authors.

## Additional information

**Competing financial interests:** The authors declare no competing financial interests.

