## [Peer Review File · Nature Communications]

Reviewers' comments:

Reviewer #1 (Remarks to the Author):

Both of my colleagues and me read this revised version, finding that the revised manuscript fully meets our requirements, so Nature Commun. can accept this nice work for publication. This comments made are based on my previous comments while the authors submitted to manuscript to Nature Materials.

Reviewer #2 (Remarks to the Author):

The revised manuscript and the authors' responses to reviewer comments retain fundamental inconsistencies.

The authors assert in the supplementary information and the their response to Reviewer #1 that PFM cannot resolve the difference between antiferroelectricity and ferroelectricity. On the contrary, it is certainly possible and in fact this is one of the unique strengths of PFM. The distinction is demonstrated in the symmetry of the piezoresponse-applied voltage cycle, the so-called "butterfly" curves like those shown in Figure S13, along with the 180° phase shift observed on reversal of the DC poling voltage. This is one of the most important features of PFM when applied to ferroelectric materials.

Neither the revised manuscript nor the authors' response address an important question relating to the the purported polarization-electric field (P-E) measurements, like those in Figures 4, 5 & S8 for compound 2, which exhibits large apparent dielectric constant, over two orders of magnitude larger than the values obtained from dielectric spectroscopy (figures S4 to S6). This is important, because the P-E measurements are in practice charge-voltage (Q-V) measurements on a capacitor, where the measured charge includes several contributions: dielectric response, polarization reversal, and free charge.

If we consider this distinction further, we must examine each P-E curve first as a Q-V curve. Consider first the contributions from conductance and space charge. Contrary to the Authors's assertions in the response, space charge can indeed mimic the symmetry of the antiferroelectric response in the case of electrically symmetric electrodes, because mechanisms like charge injection and ion accumulation have no preference for either positive and negative bias, leading to antisymmetric Q-V curves, mimicking antiferroelectricity. Conductance and space charge contributions are expected to dominate at low frequencies and are therefore more likely to show up in slow Q-V hysteresis, like that used for Figures 4, 5, & S8.

The paper is not suitable for publication in Nature communications in its present form, due not only to the problematic dielectric and hysteresis results, but also due to the unremarkable physical properties. Since the synthetic methods and crystal structure determinations are the essential core of the results, I recommend transferring the paper to a more topical journal, like JACS.

REVIEWERS' COMMENTS:

Reviewer #2 (Remarks to the Author):

The second revised manuscript has revisited the piezoresponse force microscopy results by teaming up with a new collaborator. The new results and interpretation offer a much more convincing case for the existence of ferroelectricity in two of the compound made. While this is a substantial improvement in the quality of the paper, the results are not sufficiently remarkable for publication in Nature Communications. Furthermore, the paper doesn't work as it is structured. It would work better as two topical papers, one about the crystal growth and structure, and one about the electromechanical properties of two of the compounds. I recommend Crystal Growth and Design for the former and JACS for the latter.

Engineering flexibility into piezoelectric and antiferroelectric organic crystals

Magdalena Owczarek,¹ Karl A. Hujsak,² Daniel P. Ferris,¹ Aleksandrs Prokofjevs,¹ Irena Majerz,³ Przemysław Szklarz,⁴ Haucheng Zhang,¹ Amy A. Sarjeant,¹ Charlotte L. Stern,¹ Ryszard Jakubas,⁴ Vinayak P. Dravid,^{*2} J. Fraser Stoddart,^{*1}

New title of the manuscript:

Flexible ferroelectric organic crystals

Magdalena Owczarek,¹ Karl A. Hujsak,² Daniel P. Ferris,¹ Aleksandrs Prokofjevs,¹ Irena Majerz,³ Przemysław Szklarz,⁴ Huacheng Zhang,¹ Amy A. Sarjeant,¹ Charlotte L. Stern,¹ Ryszard Jakubas,⁴ **Seungbum Hong,^{*5,6}** Vinayak P. Dravid,^{*2} J. Fraser Stoddart^{*1}

(new collaborator: ⁵Materials Science Division, Argonne National Laboratory, Lemont, IL 60439, USA. ⁶Department of Materials Science and Engineering, KAIST, Daejeon 34141, Korea.)

Point-By-Point Response to the Reviewer's Comments

Reviewer 2:

The authors assert in the supplementary information and the their response to Reviewer #1 that PFM cannot resolve the difference between antiferroelectricity and ferroelectricity. On the contrary, it is certainly possible and in fact this is one of the unique strengths of PFM. The distinction is demonstrated in the symmetry of the piezoresponse-applied voltage cycle, the so-called "butterfly" curves like those shown in Figure S13, along with the 180{degree sign} phase shift observed on reversal of the DC poling voltage. This is one of the most important features of PFM when applied to ferroelectric materials.

Neither the revised manuscript nor the authors' response address an important question relating to the the purported polarization-electric field (P-E) measurements, like those in Figures 4, 5 & S8 for compound 2, which exhibits large apparent dielectric constant, over two orders of magnitude larger than the values obtained from dielectric spectroscopy (figures S4 to S6). This is important, because the P-E measurements are in practice charge-voltage (Q-V) measurements on a capacitor, where the measured charge includes several contributions: dielectric response, polarization reversal, and free charge.

If we consider this distinction further, we must examine each P-E curve first as a Q-V curve. Consider first the contributions from conductance and space charge. Contrary to the Authors's assertions in the response, space charge can indeed mimic the symmetry of the antiferroelectric response in the case of electrically symmetric electrodes, because

mechanisms like charge injection and ion accumulation have no preference for either positive and negative bias, leading to antisymmetric Q-V curves, mimicking antiferroelectricity. Conductance and space charge contributions are expected to dominate at low frequencies and are therefore more likely to show up in slow Q-V hysteresis, like that used for Figures 4, 5, & S8.

Response: In order to answer the questions and concerns of Reviewer 2, we re-evaluated the electric properties of the materials in detail. Since questions were raised whether the antiferroelectric behavior being reported is real or is generated by artefacts, and whether the materials possess genuine ferroelectric properties, we sought the help of an expert in the PFM field, namely Dr Seungbum Hong in the Materials Science Division at the Argonne National Laboratory, to help us properly address these issues. Dr Hong agreed with all points brought up by Reviewer 2 and suggested a complete re-evaluation of the materials at hand.

Therefore, three fresh samples (two single component crystals [2 and 8] and one solid solution [2•7]) were reproduced and submitted for precise PFM measurements. In all three cases, polarization domain structures were observed, and, in two cases, the piezoresponse could be switched by applying a vertical field, which implies polarization switching characteristic of ferroelectricity. Therefore, the section describing the electric properties of the crystals has been rewritten with new Figures added.

Although attempts were made to confirm the antiferroelectric properties of the crystals reported in the original text, instrumental limitations coupled with the intrinsic small size crystals, did not allow us to produce data that we felt was reliable enough for publication at this time. We decided to remove the antiferroelectric sections from the manuscript, although, as it is mentioned in our concluding remarks, the antiferroelectric behavior is not beyond the realm of possibility and needs further investigation.

The paper is not suitable for publication in Nature communications in its present form, due not only to the problematic dielectric and hysteresis results, but also due to the unremarkable physical properties. Since the synthetic methods and crystal structure determinations are the essential core of the results, I recommend transferring the paper to a more topical journal, like JACS.

Response: As mentioned above, we completely redid the characterization of the materials to address problematic data. Although we appreciate the productive criticism of Reviewer 2, we disagree with his/her opinion on the “unremarkable physical properties” of the presented crystals. We assert that the organic, highly-crystalline, flexible, piezoelectric, and ferroelectric materials that can be produced by a simple evaporation technique, makes these systems fairly unique organo-ferroelectrics. Additionally, the synthetic methods and crystallographic data elucidate potential patterns that translate chemical composition to mechanical and electric properties in an effort to design systems with improved performance. Therefore, we believe that this new version of the manuscript is appropriate for *Nature Communications*.

Following changes were applied to the manuscript:

- The title of the manuscript has been changed from “Engineering flexibility into piezoelectric and antiferroelectric organic crystals” to “Flexible ferroelectric organic crystals”

- Dr Seungbum Hong (Materials Science Division at the Argonne National Laboratory) has been added to the author list and is one of the corresponding authors
- Introduction, abstract, and discussion have been modified to properly highlight the new results
- The section about the electric properties of these materials was rewritten
- Three new figures (4-6) have been prepared

Flexible ferroelectric organic crystals

Magdalena Owczarek,¹ Karl A. Hujsak,² Daniel P. Ferris,¹ Aleksandrs Prokofjevs,¹ Irena Majerz,³ Przemysław Szklarz,⁴ Huacheng Zhang,¹ Amy A. Sarjeant,¹ Charlotte L. Stern,¹ Ryszard Jakubas,⁴ Seungbum Hong,^{*5,6} Vinayak P. Dravid,^{*2} J. Fraser Stoddart^{*1}

Response to the Reviewer's Comments

Reviewer 2:

The second revised manuscript has revisited the piezoresponse force microscopy results by teaming up with a new collaborator. The new results and interpretation offer a much more convincing case for the existence of ferroelectricity in two of the compound made. While this is a substantial improvement in the quality of the paper, the results are not sufficiently remarkable for publication in *Nature Communications*. Furthermore, the paper doesn't work as it is structured. It would work better as two topical papers, one about the crystal growth and structure, and one about the electromechanical properties of two of the compounds. I recommend *Crystal Growth and Design* for the former and *JACS* for the latter.

Response:

Since ferroelectric and piezoelectric properties of haloimidazoles arise from the specific features of their crystal structure, in our opinion these results should not be divided into two separate papers. It is worth emphasizing that, although the phenomenon of crystal bending (either plastic or elastic) has been described in the literature, we are not aware of any electrical properties that accompany these mechanical properties. In light of these facts, as stated in our previous response to the Reviewer's comments, the successful incorporation of both desirable electrical and mechanical properties in one crystalline organic material is a unique result. Therefore, we believe that the manuscript is appropriate for *Nature Communications* and will draw a lot of attention of the *Nature Communications* readers.